# Characterization and Functional Study of *FAM49B* Reveals Its Effect on Cell Proliferation in HEK293T Cells

**DOI:** 10.3390/genes13020388

**Published:** 2022-02-21

**Authors:** Yijian Chen, Yuyan Jiang, Jihui Lao, Yankuan Zhou, Lida Su, Xiao Huang

**Affiliations:** 1Institute of Cell and Developmental Biology, College of Life Sciences, Zhejiang University, Hangzhou 310058, China; bryce@zju.edu.cn (Y.C.); 12107043@zju.edu.cn (Y.J.); 3170103036@zju.edu.cn (J.L.); 21207064@zju.edu.cn (Y.Z.); 2Neuroscience Care Unit, Second Affiliated Hospital of Zhejiang University School of Medicine, Hangzhou 310009, China; 3Key Laboratory of the Diagnosis and Treatment of Severe Trauma and Burn of Zhejiang Province, Hangzhou 310009, China

**Keywords:** *FAM49B*/*Fam49b* (CYRI-B), CCND1, cell proliferation, CRISPR/Cas9

## Abstract

*FAM49B/Fam49b* is a member of the *Fam49* (Family with sequence similarity 49) gene family, which is characterized by the conserved domain, DUF1394 (Domain of Unknown Function 1394). It has also been named CYRI-B (CYFIP related RAC1 interactor B), implicating its important function of regulating RAC1-driven cytoskeleton remolding. In this study, to further investigate its functions and mechanisms affecting cell behaviors, HEK293T cells (where *FAM49B* is highly expressed) were used to establish a *FAM49B* knockout cell line by CRISPR/Cas9 genome editing technology. Our data have clearly revealed that there are triple alleles of *FAM49B* in the genome of HEK293T cells. Meanwhile, the proliferation deficiency of the *FAM49B* KO HEK293T cell line and the significantly changed cell proliferation related gene expression profiles, such as CCND1, have been uncovered. At the same time, the existence of isoform 3 has been confirmed in HEK293T cells. Our studies have suggested that *FAM49B* may also affect cell proliferation via Cyclins, besides its influence on the cytoskeleton.

## 1. Introduction

The *Fam49* (Family with sequence similarity 49) gene family consists of three members: *FAM49A/Fam49a, FAM49B/Fam49b and Fam49c*. All three members have been characterized by a highly conserved domain, DUF1394 (Domain of Unknown Function 1394) (PDB entry 3p8c) [1]. Only *FAM49A* and *FAM49B* exist in the human genome, while *Fam49c* has only been found in fish, amphibians, reptiles, birds and some lower mammals [2,3,4]. The 3D structure of FAM49B [5,6] has revealed the high similarity to a conserved domain of CYFIP proteins, the effectors of the small GTPase RAC1 [7]. Interestingly, mouse *Fam49b* has also been reported to interact with RAC1 and play its important role in RAC1-driven cytoskeleton remolding, which leads to its new nomenclature as CYRI-B (CYFIP related RAC1 interactor B) [8,9]. FAM49B interacts with RAC1, then negatively regulates RAC1 signaling, influences cytoskeletal remodeling and finally, exerts multiple biological functions, such as weakening the process of macrocytosis, phagocytosis and cell migration [10], and negatively regulating protrusion formation through inhibiting the Scar/WAVE complex [8].

Besides, some functional studies of *FAM49B/Fam49b* have shown its indispensable role in the immune system. In flounder, in vivo knockdown *Fam49b,* which lead to the significantly increasing bacterial loads in its spleen and liver [11]. In mice, *Fam49b* has been identified as the source of a new antigen presented by Qa-1b in the absence of ERAAP [12], and plays an important role in the resistance to Salmonella infection [13]. In homo sapiens, *FAM49B* suppresses T cell activation through regulating the TCR signal transduction via modulating the actin cytoskeleton [14]. Besides, compared with healthy controls, *FAM49B* and varieties of negative regulators of inflammation are highly expressed in peripheral blood mononuclear cells of patients with multiple sclerosis [15,16].

These studies have uncovered the important role of *FAM49B/Fam49b* in both cytoskeleton remodeling and the immune system. Meanwhile, some other studies have demonstrated its potential functions in cancer progression [17] and cancer cell proliferation [18,19], hinting its different functions in various cells or tissues. In order to further figure out the functions of *FAM49B* in the human cell line and the underlying mechanisms, we have established *FAM49B* knockout (KO) HEK293T cell line using CRISPR/Cas9 genome editing technology [20]. Surprisingly, we have found the potential role of *FAM49B* in cell proliferation, contributing to a deeper understanding of its functions.

## 2. Materials and Methods

### 2.1. Cell Culture

HEK293T cells were purchased from ATCC (CRL-3216) and cultured in DMEM media (Gibco, Canoga Park, CA, USA) with 10% fetal bovine serum (Gibco, Canoga Park, CA, USA), 100 μg/mL streptomycin and 100 U/mL penicillin (Sangon Biotech, Shanghai, China) in a humidified incubator (5% CO_2_, 37 °C).

### 2.2. RNA Extraction and Reverse Transcription-PCR

Total RNA in HEK293T cells was carefully extracted by TRIzol (Invitrogen, Carlsbad, CA, USA). After that, RNA was mixed with Oligo(dT) 18 Primer, incubated in 70 °C for 10 min, cold shocked on ice for 2–3 min, and then RTase M-MLV (TaKaRa, Shiga, Japan) was used for cloning the first cDNA chain. Then, the first cDNA chain was used as a template for cloning double strand DNA (98 °C, 10 s/55 °C, 15 s/72 °C, 60 s, 30× circles) (primers forward: 5′atggggaaccttcttaaagttttgac3′, reverse: 5′gcaaatcaagacgatgctgcaataa3′); PrimeSTAR^®^ HS DNA Polymerase (TaKaRa, Shiga, Japan) was used for DNA cloning. 

### 2.3. Western Blot and Cell Immunofluorescence

Cells were lysed with RIPA buffer (50 mM Tris-HCl (pH 8.0), 150 mM NaCl, 1% Triton X-100, 0.5% sodium deoxycholate, 0.1% SDS)) containing a protease inhibitor cocktail (Roche, Basel, Switzerland). Cell debris was removed by centrifugation and proteins were separated by SDS-PAGE electrophoresis and transferred onto PVDF membrane (Millipore, Burlington, MA, USA). Then, the membrane was incubated with 1 × TBST (20 mM Tris, 150 mM NaCl, 0.1% Tween-20), adding 5% BSA for blocking non-specific sites, and incubated with the first antibody overnight at 4 °C. After washing with 1 × TBST, the membrane was then incubated with the secondary antibody for one hour at room temperature, taking GAPDH as an internal reference. Antibodies used in the Western blot: anti-FAM49B (Proteintech, Chicago, IL, USA) (1:2000 dilution), anti-GAPDH (Beyotime, Shanghai, China) (1:2000 dilution), anti-CCND1 (Abcam, Cambridge, UK) (1:1500 dilution) and HRP-labeled secondary antibody (Beyotime, Shanghai, China) (1:1000 dilution).

Clean slides were put into the cell culturing plates in advance. After incubating for 24 h, cells on slides were fixed with 4% paraformaldehyde for 10 min and washed with PBS. Then, cells were treated with 0.2% Triton X-100, washed with PBS, blocked with 1% BSA for 1 h at room temperature and incubated with FAM49B antibody (1:2000 dilution) at room temperature for one hour. Then, cells were washed with PBS and incubated with a FITC-labeled secondary antibody (Beyotime, Shanghai, China) (1:1000 dilution), protected from light. Finally, cells were washed with PBS, stained with DAPI (Beyotime, Shanghai, China) and observed using the LSM710 confocal laser-scanning microscope (Carl Zeiss, Jena, Germany). 

### 2.4. Genome Editing FAM49B in HEK293T Cells 

The genome-editing plasmid was constructed first. Two oligos were synthesized (Sangong Biotech, Shanghai, China) (5′caccgacatgcacagaccttgagc3′) (5′aaacgctcaaggtctgtgcatgtc3′) and annealed at a 10× annealing buffer (100 mM Tris-HCl (pH 8.0), 10 mM EDTA, pH 8.0, 1 M NaCl)) at 95 °C for 5 min and slowly cooled down to room temperature. Then, the duplexes were ligated to *Bbs* I linearized pX330 Vector (Addgene plasmid # 42230) by T4 DNA ligase (TaKaRa, Shiga, Japan) and then transformed into DH5α (TaKaRa, Shiga, Japan). Then, the transformed DH5α was spread on the LB solid medium (tryptone 10 g/L, yeast extract 5 g/L, NaCl 10 g/L, agar 15 g/L, pH = 7.4), with ampicillin (50 μg/mL) added and incubated at 37 °C for about 12 h. Then, the positive colonies were verified by sequencing.

When the cell coverage reached 80–90%, 2.5 μg genome editing plasmids were mixed with a 100 μL OPTI-MEM medium (Gibco, Canoga Park, CA, USA); 10 μL Lipofectamine 2000 (Invitrogen, San Diego, CA, USA) was mixed with 100 μL OPTI-MEM medium, respectively, and incubated at room temperature for 5 min. Then, these two reagents were mixed and incubated at room temperature for 20 min. Finally, the mixture was added into cells and incubated at 37 °C for 4–6 h. Transfected cells were collected and diluted to 2 × 10^5^ cells/mL, then cells were inoculated into a 96-well plate and diluted from well A1 to H8 at a two-fold dilution ratio. Single-cell clones were selected after 4–5 days.

### 2.5. Genotyping of Mutant Single-Cell Lines

Genomes of the single-cell clones were respectively extracted by genome extraction kit (Beijing ComWin Biotech, Beijing, China), and then the genomes were used as templates for cloning DNA fragments spanning targeting site (98 °C, 10 s/55 °C, 15 s/72 °C, 30 s 30× circles) (primers forward: 5′ctgtctgccagagcacagtc3′, reverse: 5′tggcaaagtggaaacattca3′); PrimeSTAR^®^ HS DNA Polymerase was used for DNA cloning. Then, the purified fragments were digested with *Sml* I endonuclease (TaKaRa, Shiga, Japan). DNA fragments cloned from KO # 7 were ligated to pGEM-T Easy Vector plasmid, transformed into DH5α and spread on the LB solid medium, adding ampicillin, IPTG and X-gal in advance, and it was incubated at 37 °C for about 12 h. Then, white colonies were verified by sequencing.

### 2.6. Cell Proliferation Assay

The 1 × 10^4^ WT and KO cells were respectively cultured in five 96-well plates, each with 5 repeated wells. CCK-8 solution (Beyotime, Shanghai, China) was added into one 96-well plate for five consecutive days and incubated for 2 h at 37 °C. The absorbance was detected at a wavelength of 450 nm. The average absorbance of 5 wells for 5 consecutive days was used to draw well growth curves.

The EdU (5′-ethynyl-2′-deoxyuridine) incorporation assay was performed using the Cell-Light™ EdU DNA Cell Proliferation Kit (Ribobio, Guangzhou, China). Cells were cultured with the EdU reagent (diluted before use) for 2 h, each with 3 repeated wells. After the standard fluorescence staining procedure, prepared specimens were visualized and analyzed using a LSM710 confocal laser-scanning microscope (Carl Zeiss, Jena, Germany). 

### 2.7. Cell Cycle Analysis

The 1 × 10^6^ WT and KO cells were respectively collected for fixation in 70% ethanol, gently pipetted to mix and left overnight at 4 °C. After fixation, cells were washed with PBS and centrifuged to remove the supernatant. Then, 0.5 mL of PI (Beyotime, Shanghai, China) staining solution was added into cells, slowly and fully resuspended the cell pellet, incubated for 30 min at 37 °C and protected from light. CytoFLEX (Beckman Coulter, Brea, CA, USA) was used to analyze the proportion of cells in different periods.

### 2.8. Apoptosis Detection

The 1 × 10^5^ WT and KO cells were respectively collected by centrifugation, and the cell pellet was resuspended in 200 μL Annexin V-FITC binding solution (Beyotime, Shanghai, China), then mixed with 10 μL PI staining solution, incubated at room temperature for 15 min and protected from light. Finally, the apoptosis rate was analyzed by CytoFLEX.

### 2.9. RNA-Seq and Data Analysis

Total RNA was extracted with TRIzol. RNA-seq was performed on the Illumina HiSeq2000 platform. The significantly changed genes (FDR ≤ 0.05, log2FC ≥ 1) were used for GO analysis with DAVID online tools (https://david.ncifcrf.gov/tools.jsp, accessed on 5 December 2021) and KEGG pathway enrichment analysis (http://www.genome.jp/kegg/, accessed on 5 December 2021). RNA was reverse transcribed to cDNA (refer to Section 2.2) and real-time quantitative PCR was conducted in triplicate by CFX Connect Real-time System (Beckman Coulter, Brea, CA, USA) with iQ™ SYBR^®^ Green super mix (Beckman Coulter, Brea, CA, USA), taking GAPDH as the internal reference. The ΔΔCT method was used to determine relative mRNA levels. 

### 2.10. Statistical Analysis

Experiments were replicated at least three times (except for cell cycle assay and apoptosis detection). The statistical significance of differences between the studied groups was analyzed using the Student’s *t*-test. Data were presented as the mean ± SD (standard deviation). The *p* value less than 0.05 was considered to be statistically significant. Prism GraphPad Software 5.0 (GraphPad Software Inc., San Diego, CA, USA) was used for statistical analysis.

## 3. Results

### 3.1. Characterization of FAM49B in HEK293T Cells

*FAM49B*, one of the members of the *Fam49* gene family, has been characterized by the conserved domain, DUF1394 [1]. Recently, the biological functions and mechanisms of *FAM49B/Fam49b* have been revealed [5,8,10], but few studies have clearly described the characteristics of *FAM49B*. We have sorted out the latest information about human *FAM49B* in the database (https://www.ncbi.nlm.nih.gov/gene/51571, accessed on 28 December 2021), and have described its gene structure and expression (Figure 1A) in detail. 

*FAM49B* contains 23 exons and 22 introns. A total of 78 transcripts and four protein isoforms of this gene have been recorded. Except for isoform 4, which is translated by only one transcript, the other three isoforms are translated from 29 transcripts, 34 transcripts and 14 transcripts, respectively. Isoform 1 and isoform 3 are both composed of 324 amino acids (aa), while isoform 4 is composed of 316 aa; except for the difference in the first 25 aa, the downstream aa sequences of the isoform 1, isoform 3 and isoform 4 are completely the same. The molecular weights of the four isoforms are, respectively, about 36.7 kDa, 20 kDa, 36.8 kDa and 35.8 kDa.

The genome-wide RNA expression profiles of protein-coding genes in human cell lines have demonstrated the high expression of *FAM49B* in HEK293T cells (https://www.proteinatlas.org/, accessed on 25 December 2021). Thus, we re-verified the expression of *FAM49B* in HEK293T cells at both RNA and protein levels by RT-PCR (Figure 1B) and Western blot (Figure 1C). Further, we studied the cellular localization of FAM49B in HEK293T cells by immunofluorescence. It has demonstrated that FAM49B is widely distributed both in the cytoplasm and nucleus (Figure 1D). In summary, our studies have confirmed the expression of *FAM49B* in HEK293T cells.

### 3.2. Establishing FAM49B Mutant HEK293T Cell Line

To study the functions of *FAM49B* in HEK293T cells, we undertook establishing the *FAM49B* knockout (KO) HEK293T cell line. Firstly, we designed single-guide RNA (sgRNA) to KO *FAM49B* at the targeting site (T1). Then, the genomes of HEK293T cells were extracted for genotyping. DNA fragments (316 bp) spanning T1 were cloned from extracted genomes for sequencing. The sequencing results demonstrating messy peaks appeared on the DSB site, suggesting the genomes of the mutant cells had been edited (Figure 2A). 

The target site T1 contains the recognition sequence of endonuclease *Sml* I; this recognition sequence could be broken by mutations of different types. We digested the cloned fragments spanning T1 with *Sml* I endonuclease and observed that there were still some cloned fragments in the mutant group that were not digested (~316 bp), while all the cloned fragments in the wildtype (WT) group were digested into two fragments (~197 bp and ~119 bp), indicating the target site was mutant (Figure 2B). Then, mutant cells were limitedly diluted into 96-well plates, and then 21 single-cell lines were obtained and analyzed by *Sml* I digestion (Figure 2C). Finally, we obtained five mutant single-cell lines, # 7, # 11, # 15, # 16 and # 17. Then, the Western blot experiment was conducted to examine the expression of *FAM49B* in mutant single-cell lines (Figure 2D). It demonstrated that the expression of *FAM49B* in the mutant single-cell line # 7 (KO # 7) was the least, and almost undetectable. Those results have suggested that *FAM49B* is successfully mutant in KO # 7.

### 3.3. Triple Alleles of FAM49B Exist in HEK293T Genome

To further clarify the specific mutation information about the target site, DNA fragments (316 bp) spanning T1 in KO # 7 were cloned for genotyping. The sequencing results demonstrated that there were three types of mutations (delete 22 bp, insert 1 bp and insert 38 bp) in KO # 7 (Figure 3A). The electrophoresis result demonstrated three bands of the expected size (~294 bp, ~317 bp and ~354 bp) (Figure 3B), consistent with the sequencing results. To reconfirm that the KO # 7 derived from a single colony, we performed a further limiting dilution of KO # 7. As a result, we obtained 10 single-cell clones, which were then tested by *Sml* I digestion. The cloned sequences spanning T1 of those 10 clones were not digested and appeared with the same band pattern as KO # 7, while the cloned fragments in the WT group were all digested into two fragments (~197 bp and ~119 bp) (Figure 3C). It further demonstrated that the KO # 7 was a single-cell line, demonstrating that triple alleles of *FAM49B* exist in the genome of HEK293T cells, and all of them have been mutant by genome editing.

### 3.4. FAM49B KO Cells Exhibit Proliferation Deficiency

During cell culturing, we surprisingly found that KO # 7 exhibited remarkable proliferation deficiency, hinting at the connection between *FAM49B* and cell proliferation. Firstly, through the records of cell growth for 5 consecutive days, we observed that KO cells grew slower than WT cells (Figure 4A). Subsequently, we tested the proliferation capability of WT and KO cells. WT and KO cells of the same concentration (1 × 10^4^ cells/well) were respectively seeded into five 96-well plates for a CCK8 assay. A growth curve drawn by the average absorbance (proportional to the number of cells) demonstrated that the number of WT and KO cells was approximately equal on the first day. From the second day to the fifth day, KO cells were less than WT cells, and the difference became more obvious by days (Figure 4B). Concurrently, the EdU incorporation assay demonstrated that the EdU incorporating proportion (proportional to the cell proliferation rate) of KO cells was obviously lower than that of WT cells (Figure 4C,D). In short, these results demonstrated that KO # 7 exhibited proliferation deficiency.

### 3.5. FAM49B Mutant HEK293T Cells Display No Remarkable Differences in Cell Cycle Arrests or Apoptosis

The regulation of cell proliferation is very complex, related to many biological processes, such as cell cycle arrests, apoptosis, differentiation and extracellular matrix, etc. [21,22,23,24,25]. Then, we studied the cell cycle and apoptosis of KO # 7. The cell cycle analysis, based on the distribution of DNA content, demonstrated the average proportions of WT cells in the three stages (G0/G1, S, G2/M) are 39.432%, 48.268% and 12.300%, respectively, and the KO cells are 36.195%, 50.813%, and 12.992%, respectively (Figure 5A,B). Concurrently, the apoptosis analysis by the Annexin V-FITC/PI double staining demonstrated that the average apoptosis rate of WT and KO cells were 1.60% and 1.36%, respectively (Figure 5C,D). Those two detections demonstrated no remarkable difference in the proportions of different stages of cell cycle or apoptosis rates between WT and KO cells, suggesting that the proliferation deficiency of *FAM49B* KO HEK293T cells is unrelated to cell cycle arrests or apoptosis.

### 3.6. The Alternative Splicing of FAM49B in HEK293T

We wondered why cell proliferation slowed down but cell cycle or apoptosis did not change significantly. Thus, we re-verified the expression of *FAM49B* in KO # 7. Total RNA was extracted as the template for RT-PCR (Figure 6A) and DNA fragments (400 bp) spanning T1 were cloned for TA cloning. Then, transcript colonies No. 6, 7, 8, and 9 were selected for sequencing at the T1 site. By sequence alignment analysis with the corresponding sequence in the WT, we found that the cloned sequence in colony No. 6 and No. 8 lost the targeting exon 14, becoming 317 bp (Figure 6C). The sequence in No. 7 inserted one base, becoming 401 bp (Figure 6D). The sequence in No. 9 had no difference in length but had many mismatched bases with the WT (Figure 6E). Interestingly, this transcript completely matched the sequence of exon 12. 

According to our characterization of *FAM49B* (Figure 1A), the start codon of transcripts encoding isoform 3 is located on exon 12 and the mutations on exon 14 would not influence the expression of isoform 3. Thus, we optimized the Western blot experiment (all four isoforms could be detected by FAM49B polyclonal antibody) and found a small amount of FAM49B (~37 kDa) still existed in KO # 7 (Figure 6B). According the genotyping and the second time limiting dilution results, the triple alleles of *FAM49B* in KO # 7 were completely mutant. Therefore, we reconfirmed that the isoform 3 and its encoding transcripts of *FAM49B* actually exist in HEK293T cells.

### 3.7. CCND1 Expression Is Dramatically Decreased in FAM49B Mutant Cells

To clarify the specific mechanism of *FAM49B* affecting cell proliferation, we conducted transcriptome analysis by RNA-Seq. It demonstrated that the gene expression profiles were significantly changed between WT and KO cells. A total of 19,938 transcripts were overlapped in WT and KO cells, while 2017 transcripts and 2150 transcripts were expressed in WT or KO cells, respectively (Figure 6A and Appendix A). Concurrently, differentially expressed genes (counted to 15,457) were visualized with a volcano map. The red dots represented significantly up-regulated genes (counted to 209) and the blue dots represented significantly down-regulated genes (counted to 109) (Figure 6B and Appendix A). Meanwhile, 318 significantly expressed genes performed a cluster analysis of expression patterns. This demonstrated that at least five of the ten types of expression patterns were quite different (Δlog2(fpkm + 1) ≥ 0.5)) (Figure 6C).

Furthermore, we performed GO (gene ontology) enrichment analysis on significantly differentially expressed genes between WT and KO # 7 (Figure 6D and Appendix A). This demonstrated that the expression of gene groups regulating cell proliferation and epithelial cell proliferation were significantly down-regulated in *FAM49B* KO HEK293T cells. It also confirmed that *FAM49B* was essential for cell proliferation. Notably, compared with that in the WT group, the expression of CCND1(Cyclin D1), a gene important for cell proliferation and cell cycle transition in KO # 7, was significantly down-regulated, which was confirmed by the real-time quantitative PCR analysis (Figure 6E) and Western blot (Figure 6F). These results suggested that *FAM49B* might regulate cell proliferation by promoting the expression of CCND1 in HEK293T cells (Figure 6G). Besides, among the down-regulated genes, we also noticed that gene groups related to organ development and cell differentiation were significantly down-regulated, such as bone development, heart development, epithelial cell differentiation and keratinocyte cell differentiation, etc. Among the up-regulated genes in HEK293T KO cells, the gene groups related to the sensory perception of sound and mechanical stimulus, ear and inner ear morphogenesis and ear and inner ear development were significantly up-regulated. In conclusion, GO enrichment analysis suggested that *FAM49B* was not only related to cell proliferation but also may influence organ development, cell differentiation and regulate ear morphogenesis and sensory perception of sound.

## 4. Discussion

Functional studies have revealed the important function of *FAM49B/Fam49b* on cytoskeletal remodeling [6,8,9,10,11,14] and immune system in varieties of species [11,12,13,15,16]. However, several studies also indicated the potential role of *FAM49B* involving cell proliferation [18,19]. Based on these findings, we tried to investigate the underlying mechanisms by establishing a *FAM49B* KO HEK2793T cell line. Through CRISPR/Cas9 genome editing technology, we successfully introduced indels on the target loci (Figure 2). Interestingly, our data also clearly revealed that there were three alleles of *FAM49B* in the genome of HEK293T. This is consistent with the cytogenetic analysis of HEK293T cell line (Bylund et al., 2004; Lin et al., 2014) since HEK293T is a hypotriploid human cell line. According to ATCC information (ATCC; CRL-3216), the modal chromosome number of HEK293T is 64, occurring in 30% of cells, and the rate of cells with higher ploidies is 4.2%. After the second limiting dilution and genotyping, we finally confirmed three mutant alleles of *FAM49B* in KO # 7 (Figure 3).

Subsequently, we found the proliferation deficiency of tri-loci *FAM49B* KO HEK293T cells compared with the WT cells, which was further confirmed by the CCK8 assay and EdU incorporation assay, respectively (Figure 4). Concurrently, no remarkable differences in the cell cycle or apoptosis between WT and KO cells were detected (Figure 5). Unfortunately, the *FAM49B* KO cell line was not properly preserved, so we were not able to overexpress *FAM49B* in the *FAM49B* KO cells to finally verify whether the cell proliferation defect could be restored. 

We wondered about the underlying mechanisms causing the cell proliferation deficiency but hardly affecting the cell cycle arrests or apoptosis of *FAM49B* KO HEK293T. Subsequently, we re-verified the expression of *FAM49B* at the RNA and protein levels. Through RT-PCR and TA cloning experiments, we confirmed the existence of the FAM49B isoform3 and its encoding mRNA. The sgRNA target site T1 is located on exon 14, which could destroy the expression of isoform 1, but has no effect on the expression of isoform 3 due to alternative splicing (Figure 6).

In order to clarify the specific mechanism of *FAM49B* affecting cell proliferation, we conduct transcriptome analysis in WT and KO cells by RNA-Seq. The significantly changed gene expression profiles demonstrated the down-regulation of cell proliferation related genes, such as CCND1 (Figure 7). As a crucial member of the RB (retinoblastoma) pathway, CCND1 enables E2F transcription factors to activate genes required for entry into the S phase by phosphorylating Rb [26,27,28]. However, HEK293 was derived from human embryonic kidney cells by transfection with adenovirus (Ad5) DNA expressing E1A [29]. E1A combines and inactivates the Rb family, resulting in the activation of E2F and the transition from the G1 to S phase [30,31,32] (Figure 7E). Therefore, we found the *FAM49B* KO HEK293T cells were not restricted to the G1/S checkpoint, but the proliferation rate slowed down. Our studies suggest that *FAM49B* may influence cell proliferation by regulating the expression of CCND1 in a cytoskeleton-independent way. Besides, GO enrichment analysis demonstrated the potential connection between *FAM49B* with organ development and cell differentiation, especially ear development and morphogenesis, bone development and epithelial cell differentiation, and the specific connection and underlying mechanisms need further studies to clarify.

However, the cytoskeleton and cell cycle are inextricably linked in eukaryotic cells [33,34]. During mitosis, the cytoarchitecture changes with the cell cycle progression [35,36]. *FAM49B* has been reported to regulate cytoskeleton remodeling by combining with RAC1 [8,9,10]. KEGG pathway enrichment analysis of differential genes (Kyoto Encyclopedia of Genes and Genomes) demonstrated the significantly changed gene groups related to focal adhesion and the ECM-receptor interaction (Appendix A). The ‘adhesion checkpoint’ hypothesis [37] suggests that cell-cycle-dependent changes in adhesion complexes and the cytoskeleton are essential for cell-cycle progression and division, suggesting that *FAM49B* also may influence cell proliferation in a cytoskeleton-dependent way. Whether *FAM49B* affects cell proliferation in cytoskeleton dependent or independent manners remains to be resolved.

## 5. Conclusions

Our data have clearly revealed triple alleles of *FAM49B* in the genome of HEK293T cells. Meanwhile, we have found the proliferation deficiency of the *FAM49B* KO HEK293T cell line. Further, the isoform 3 of FAM49B has been confirmed in HEK293T cells. RNA-Seq and data analysis of differentially expressed genes demonstrated the significantly changed expression of gene groups related to cell proliferation. Our studies have suggested that *FAM49B* may affect cell proliferation via CCND1, besides its influences on the cytoskeleton.

## Figures and Tables

**Figure 1 genes-13-00388-f001:**
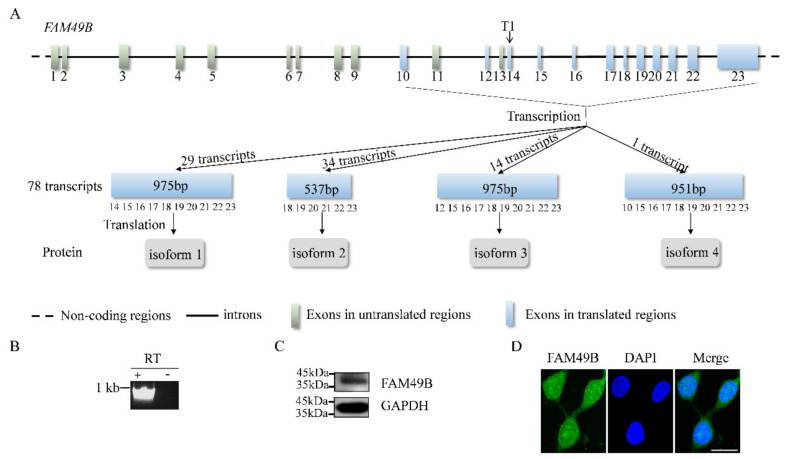
Characterization of human *FAM49B*. (**A**) The *FAM49B* contains 23 exons (highlight with blue rectangles) and 22 introns (highlight with the solid black line); non-coding regions are highlighted with the black dashed line. T1 symbol of sgRNA target site is on the exon 14. Four protein isoforms are highlighted with gray rectangles. (**B**) Agarose electrophoresis is the result of RT-PCR (reverse transcription PCR) products of *FAM49B* in HEK293T cells and the control group directly used the extracted total RNA as the template for the PCR cloning. (**C**) Western blot result of FAM49B in HEK293T cells with GAPDH as an internal reference and the molecular weight of FAM49B and GAPDH is about 36.7 kDa and 40 kDa, respectively. (**D**) Immunofluorescence staining of FAM49B in HEK293T cells, scale bar, 20 μm.

**Figure 2 genes-13-00388-f002:**
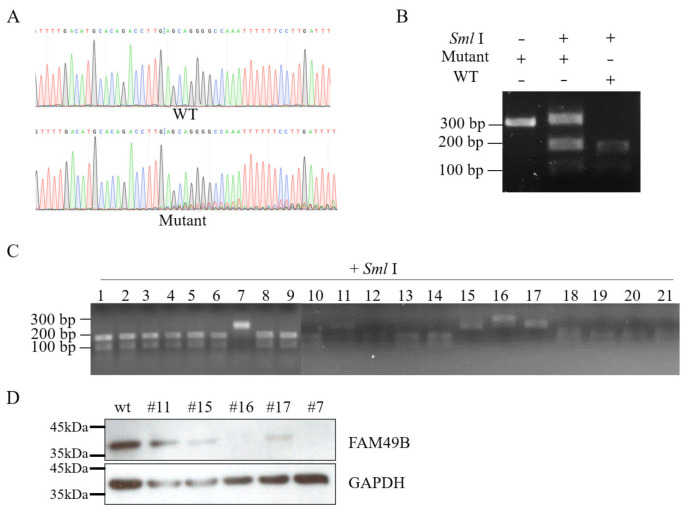
CRISPR/Cas9 genome editing to establish *FAM49B* KO HEK293T cell line. (**A**) Sanger sequencing of the target site. Messy peaks represented various mutant types. WT: wildtype HEK293T cells; mutant: *FAM49B* mutant HEK293T cells. (**B**) The DNA fragments spanning the target site (T1) cloned from WT genome were totally cut into 119 bp and 197 bp by *Sml* I. The DNA fragments cloned from the mutant genome were partially digested by *Sml* I. (**C**) DNA fragments cloned from 21 single-cell clones were testified by *Sml* I digestion. The DNA fragments cloned from single-cell clones, # 7, # 11, # 15, # 16 and # 17 were nearly unbroken after *Sml* I digestion. (**D**) Western blot result of five single-cell clones, # 7, # 11, # 15, # 16 and # 17. Almost no FAM49B was detected in KO # 7.

**Figure 3 genes-13-00388-f003:**
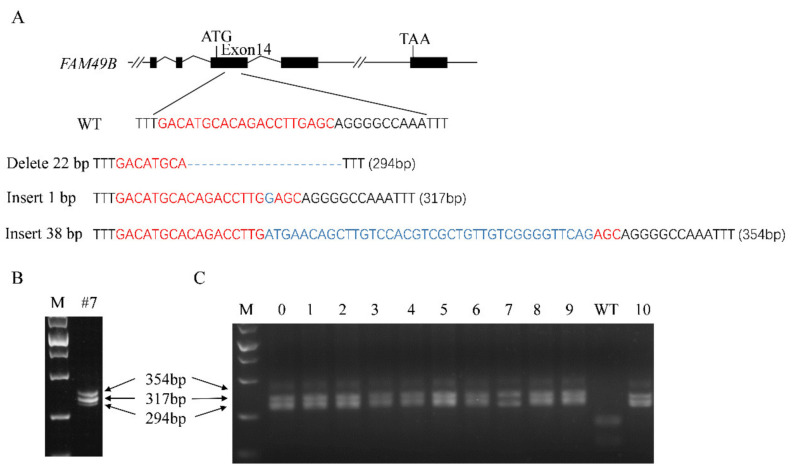
Triple alleles of *FAM49B* exist in HEK293T genome. (**A**) Genotyping demonstrated that there were three types of mutations at the target site in KO # 7. Mutant 1 (with 22 bp deletions), mutant 2 (with 1 bp insertion) and mutant 3 (with 38 bp insertions); the target site was highlighted with red and mutant bases were highlighted with blue. (**B**) Agarose gel electrophoresis was the result of DNA fragments spanning T1 in KO # 7, consistent with sequencing results. (**C**) DNA fragments spanning T1 were cloned from newly selected 10 single-cell clones and digested with *Sml* I. DNA fragments cloned from WT were cut into 119 bp and 197 bp. DNA fragments cloned from 10 single cell clones were unbroken, consistent with the cloned sequence from KO # 7.

**Figure 4 genes-13-00388-f004:**
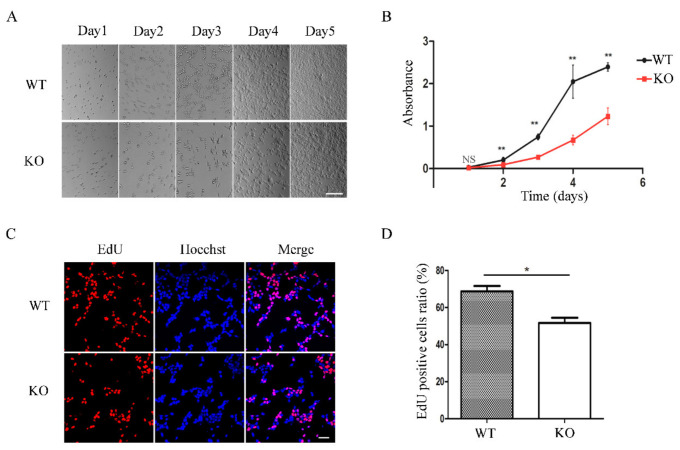
*FAM49B* mutant cells exhibited proliferation deficiency. (**A**) The growth of the WT and KO cells was recorded for five consecutive days. The initial concentration was 5.5 × 10^4^ cells per well, repeated 3 times. Scale bar, 200 μm (**B**) The growth curve of WT and KO cells was drawn by CCK-8 experiment, repeated 3 times and the data were presented as mean ± SD. NS—no significance, ** *p* < 0.01 (Student’s *t*-test). (**C**) EdU incorporation assay of relative Hoechst-stained cells and EdU-incorporated cells. Scale bar, 50 μm (**D**) Different EdU positive ratio between WT and KO cells, repeated three times and the data were presented as mean ± SD. * *p* < 0.05 (Student’s *t*-test).

**Figure 5 genes-13-00388-f005:**
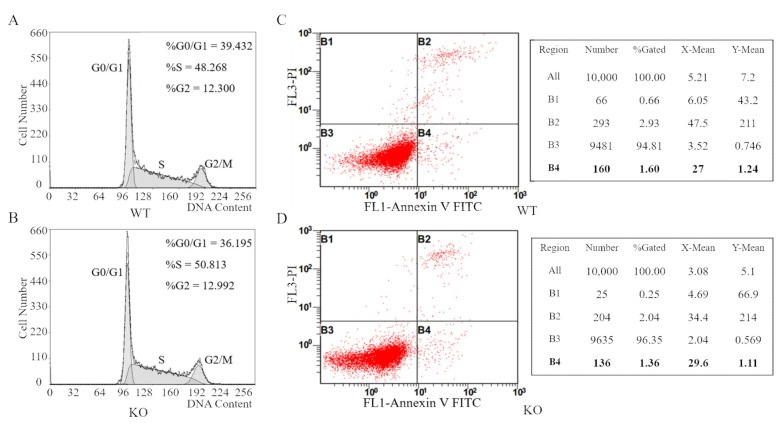
*FAM49B* KO HEK293T cells displayed no remarkable differences in cell cycle arrests or apoptosis. (**A**,**B**) The proportions of WT and KO cells in different stages of the cell cycle, counted by flow cytometry. No remarkable difference was detected. (**C**,**D**) The proportions of WT and KO cells in apoptosis counted by flow cytometry. B2, B3 and B4 represented necrotic cells, normal live cells and early apoptotic cells (highlight with bold font), respectively. B1 was the detection error within the allowable range.

**Figure 6 genes-13-00388-f006:**
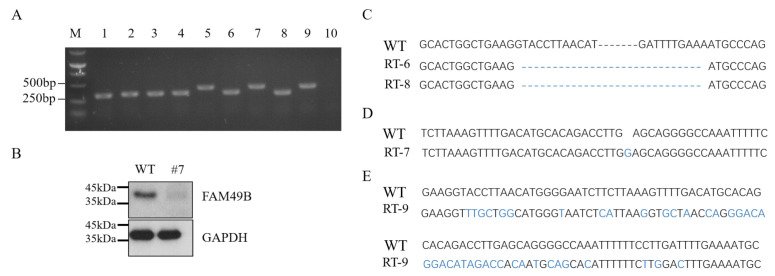
Isoform 3 of FAM49B exists in HEK293T cells. (**A**) Agarose electrophoresis results of nine single colonies after TA cloning of RT-PCR products, colony 10 as negative control. (**B**) Western blot result of FAM49B in KO # 7, GAPDH as internal reference. A small amount of FAM49B was detected. (**C**–**E**) Comparison of the transcripts between WT with colony RT-6 (transcript 6), RT-7, RT-8 and RT-9, respectively. Exon 14 (the targeting exon) was lost in RT-6 and RT-8, one base was inserted in RT-7 and many mismatched bases existed in RT-9. Differences with WT sequence were highlighted with blue.

**Figure 7 genes-13-00388-f007:**
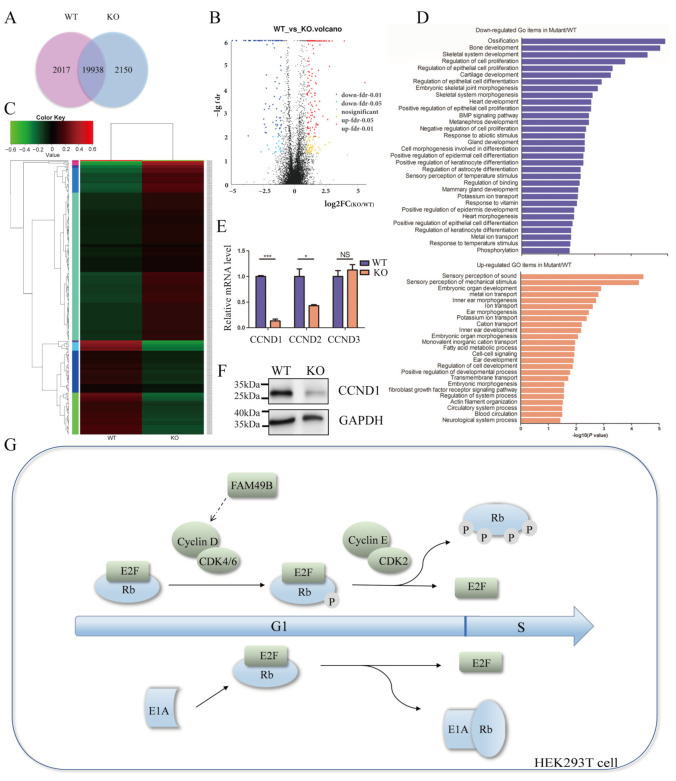
The expression of CCND1 is dramatically decreased in *FAM49B* KO cells. (**A**) Total genes nominated in a Venn diagram. A total of 19,388 genes were detected in both two samples. (**B**) Visualized volcano map of differentially expressed genes of two samples. The abscissa is the logarithm of the fold change value of the difference in gene or transcript expression between two samples, and the ordinate is the logarithm of the statistical test value of the difference in gene or transcript expression. The smaller the FDR-value, the greater the difference in expression. (**C**) The heat map demonstrated the hierarchical clustering of differentially expressed genes (>2-fold change; FDR < 0.05). (**D**) Bar plot of the GO analysis of the biological processes between WT and KO cells. The significantly changed genes (FDR ≤ 0.05, log2FC ≥ 1) were used for GO analysis. The y axis showed the GO terms, and the x axis showed the enrichment significance *p* values for these top enriched GO terms (*p* < 0.05). (**E**) Confirmation of reduced expression of CCND genes by quantitative RT-PCR. Data were normalized to GAPDH. Three parallel experiments in each group were conducted. Data were presented as mean ± SD. NS—not significant, * *p* < 0.05, *** *p* < 0.001 (Student’s *t*-test). (**F**) Total proteins were extracted from WT and KO cells and subjected to Western blot analysis of CCND1. (**G**) The role of *FAM49B* in cell cycle transition. *FAM49B* promotes the expression of cyclin D so as to phosphorylate Rb and eliminates the inhibition of E2F by Rb. E1A could compete with Rb for E2F then directly activate E2F, leading to immortality of HEK293T.

## Data Availability

All data supporting reported results can be found in Appendix A.

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
