# Peer review of "Characterization and Functional Study of FAM49B Reveals Its Effect on Cell Proliferation in HEK293T Cells"

_genes, 2022, doi:10.3390/genes13020388_

Round 1
Reviewer 1 Report
Chen et al. report a study about the still largely un-characterized FAM49B gene in a human embryo kidney cell line. The authors generated a FAM49B knockout cell line using CRISPR/Cas9 genome editing. The authors succeed apparently in knocking out all three of the supposed alleles of FAM49B in the HEK293T cell line, and report that FAM49B loss-of-function affects mainly cell proliferation whilst not changing significantly the duration of the cell cycle phases, despite a reduction in CCND1 expression. The authors also claim to have confirmed FAM49B isoform 3 existence and expression in HEK293T cells.
This manuscript is potentially interesting, however, a few issues have to be clarified:
-It is not clear which of the 4 isoforms of FAM49B the anti-FAM49B antibody can detect. The authors should preferably use antibodies capable of distinguishing the various isoforms or at least state in the Results section which isoforms are detected by the antibody used.
-FAM49B should be localized on human Chr. 8. The authors should show a karyotype of FAM49B KO cells to conclusively demonstrate that three copies of FAM49B exist in HEK293T cells.
-To obtain triple KO of all FAM49B alleles in HEK293T cells the authors resorted to cloning of cells by limiting dilution. This of course exposes to the risk of selecting other mutations within the cell genome that could cause a reduction in cell proliferation. The authors should demonstrate that the observed reduction in cell proliferation is due primarily to FAM49B loss-of- function by performing rescue experiments. The authors should overexpress FAM49B in FAM49B KO HEK293T cells to show that the proliferation rate of wt HEK293T cells can be restored. This could be also a good opportunity to test the various isoforms for their possible specific role in controlling cell proliferation.
Author Response
Thank you for reviewing this manuscript and providing us so many constructive comments that are very helpful to us.
Response 1:
This advice is very important. We did not state in the manuscript exactly which isoforms the FAM49B antibody could detect. In fact, the Fam49B polyclonal antibody was commercially purchased from Proteintech™. According to the product sheet, it was obtained by immunizing rabbits with the full-length FAM49B isoform 1. Thus, all four protein isoforms can be detected potentially. So, it cannot be used to distinguish the various isoforms. We have revised our manuscripts and stated this in Result Section 3.6.
Response 2:
The karyotype studies on FAM49B KO cells are indeed the most convincing evidence for demonstrating three copies of FAM49B exist in HEK293T cells. But combining the characteristics of HEK293T described in ATCC and three types of mutations we found in the FAM49B KO cell line, we believe it is sufficient to demonstrate the triple alleles of FAM49B in HEK293T genome.
Response 3:
Thank you for your professional and very helpful suggestion. The rescue experiments are definitely necessary to prove the specificity of the KO phenotype. We had planned to carry out this experiment, but unfortunately, due to the sudden pandemic of COVID-19 and quarantine, all the cell lines stored in the laboratory died because of the exhaustion of the liquid nitrogen. To complete this experiment, we need to re-establish the cell lines, which could expend a relatively long period, and currently we cannot finish it in a short time.
Reviewer 2 Report
In their manuscript „Characterization and functional study of FAM49B reveal its cell proliferation role in HEK293T cells”, Chen et al. investigate the role of FAM49B/Fam49b in cell proliferation of HEK293T by establishing a novel mutant with downregulated FAM49B via a CRISPR/Cas9 approach. The manuscript provides enough evidence, but there are some minor issues to be resolved.
The manuscript is full of spelling errors, typos, wrong use of time etc. It should be revised by a native speaker.
In the method sections, some important parts are missing, mainly: Information on the used concentrations of antibodies for western blots and immunostaining, information on fluorescence microscopy imaging, information on statistics (e.g. filters, objective, imaging media, …). An important missing part is about the number of experiments done (e.g. repeats of same experiments) and statistical analysis.
The information in Figure 4 and 5 needs to be supported by statistical analysis and information on the number of experiments.
The authors show that knockout of FAM49B leads to proliferation deficiency, while cell cycle stage and apoptosis appear to not be influenced. The authors make multiple comments about the interaction of the protein with RAC1 and its influence on cytoskeleton structure. Additional experiments about cell migration, status of the cytoskeleton, and invasion would have been interesting to be performed.
The authors show that Cyclin D1 is significantly downregulated in the knockdown and show their results of RNA sequencing analysis. They should discuss in a bit more detail the upregulated/downregulated genes that they think are of interest for the current publication, mainly those for cell cycle and cytoskeleton structure. It looks like epithelial cell cycle regulation and epithelial cell differentiation are two of the gene groups that are most downregulated, while embryonic developmental genes appear to be upregulated. Does this overlap with the hypothesis of the authors that the studied gene/protein interacts with the cytoskeleton? Do the authors see any other signs of cell dedifferentiation?
The discussion is mostly a summary of the results and in my opinion should be focused a bit more on discussing the present results in the face of the current literature.
Author Response
Response: Thank you for reviewing this manuscript and providing us so many constructive comments that are very helpful to us.
Response 1:
We have invited one native speaker, who is also a faculty member in our school, to do proofreading and help us revise the manuscript. Unfortunately, he’s currently unavailable in such a short time to complete the job. However, our invitation still keeps when he has the time. And the authors have tried to revise the manuscript on the language issue at our best. We think this version of the manuscript is much better than the first one.
Response 2:
Thank you for your professional suggestion. In Method Section 2.3, we have added information on the used concentrations of antibodies for western blots and immunostaining, information on fluorescence microscopy imaging, information on statistics. In Method Section 2.6, we have added information on the number of experimental repetitions.
Response 3:
Thank you for your professional suggestion. For CCK-8 and EdU cell proliferation assays (Fig 4), we have described our parallel repeat experiment in the Method Section 2.6. After doing the first cell cycle and apoptotic experiments, we were going to repeat those experiments twice again. But unfortunately, due to the sudden pandemic of COVID-19 and quarantine, all the cell lines stored in the laboratory died because of the exhaustion of the liquid nitrogen. To complete this experiment, we need to re-establish the cell lines, which could expend a relatively long period, and currently we cannot finish it in a short time. We have highlighted this in the Discussion section.
Response 4:
Thank you for your professional suggestion. It's a very interesting topic to confirm the reported function and mechanisms of Fam49b in different cell lines. However, HEK293T cells are not well models for studying cell migration or invasion with significance. Actually, we do investigate the functions of FAM49B in cancer cell lines, such as U251 Human glioma cells, and found a very interesting phenomenon that FAM49B KO U251 cells exhibit increase on cell migration and invasion, which is consistent with its RAC1 dependent cytoskeleton function. However, our manuscript here is highlighting the effect of FAM49B on cell proliferation, and we will discuss this topic in other context in future.
Response 5:
Thank you for your professional suggestion and detailed interpretation on our data. We have re-analyzed the GO enrichment results. We have found that FAM49B KO led to significantly differentially expressed gene groups also associated with embryonic development and cell differentiation besides cell proliferation, and we have highlighted this in Result Section 3.7. We do perform an investigation on the function of Fam49b in vivo systems due to its temporal and spatial expression during Xenopus tropicalis embryogenesis, which could possibly reveal its potential functions and mechanisms on development.
Response 6:
Thank you for your professional suggestion. We have modified the discussion section to incorporate data from KEGG signaling pathway analysis and found that FAM49B KO has a significant impact on the cytoskeleton, particularly focal adhesion and ECM-receptor interactions. Combined with the ‘adhesion checkpoint’ hypothesis, we suggest that FAM49B may influence cell proliferation by affecting the cytoskeleton.
Thank you again for your patience and professional suggestions on this manuscript!